# Days-in-Milk and Parity Affected Serum Biochemical Parameters and Hormone Profiles in Mid-Lactation Holstein Cows

**DOI:** 10.3390/ani9050230

**Published:** 2019-05-10

**Authors:** Xuehui Wu, Hui-Zeng Sun, Mingyuan Xue, Diming Wang, Leluo Guan, Jianxin Liu

**Affiliations:** 1Institute of Dairy Science, College of Animal Sciences, Zhejiang University, Hangzhou 310058, China; xuehuiwu@zju.edu.cn (X.W.); huizeng@ualberta.ca (H.-Z.S.); myxue@zju.edu.cn (M.X.); wdm@zju.edu.cn (D.W.); 2Department of Agricultural, Food & Nutritional Science, University of Alberta, Edmonton, AB T6G 2P5, Canada

**Keywords:** serum biochemical parameter, serum hormone, days-in-milk, parity, dairy cows

## Abstract

**Simple Summary:**

Serum biochemical parameters and hormones play a role in directly reflecting the physiological state and modulating the milk performance of dairy cows. However, the variability of the serum biochemical parameters and hormones in multiparous mid-lactation cows have not drawn much attention due to their supposedly stable states, especially for those cows under the same nutrition and management condition. The aim of this study was to evaluate the effects of days-in-milk (DIM, within the mid-lactation) and parity (ranging from 2–6) on serum biochemical parameters and hormone profiles based on a large cohort of dairy cows. The results showed that DIM and parity contribute to the variations in serum biochemical parameters and hormones related to protein status, energy supply, liver and kidney function, and oxidative stress of mid-lactation dairy cows, with the effect of DIM being dominant over parity. Our result suggested that the DIM periods and parity should be taken into consideration to optimize nutritional strategies in order to improve the milk performance traits more precisely.

**Abstract:**

It is well known that serum biochemical parameters and hormones contribute greatly to the physiological and metabolic status of dairy cows. However, few studies have focused on the variation of these serum parameters in multiparous mid-lactation cows without the interference of diet and management. A total of 287 Holstein dairy cows fed the same diet and maintained under the same management regime were selected from a commercial dairy farm to evaluate the effects of days-in-milk (DIM) and parity on serum biochemical parameters and hormone profiles. Milk yield and milk protein content were affected by DIM and parity (*p* < 0.05). Milk protein yield showed a numerically decreasing trend with parity, and it was relatively constant in cows with parities between 2 and 4 but lower in cows with parity 6 (*p* = 0.020). Ten and five serum biochemical parameters related to protein status, energy metabolism, liver and kidney function, and oxidative stress were affected by DIM and parity, respectively (*p* < 0.05). Glucagon, insulin-like growth factor 1 concentration, and the revised quantitative insulin sensitivity check index were significantly different (*p* < 0.05) among cows with different DIM. Parity had no effect on hormone concentrations. An interaction between DIM and parity effect was only detected for glucagon concentration (*p* = 0.015), which showed a significantly increasing trend with DIM and overall decreasing trend with parity. In summary, DIM and parity played an important role in affecting the serum biochemical parameters and/or hormones of dairy cows, with serum parameters affected more by DIM than parity.

## 1. Introduction

Serum biochemical parameters and hormones directly reflect the physiological state of dairy cows [1] and may play a role in modulating milk production [2]. However, most studies focusing on factors affecting milk production have been based on differences in nutrition [2], management [3], or health [4], and only used a small number of cows. Recent studies have reported that healthy cows can have different milk production levels even when they are fed the same diet and managed under the same conditions [5]. However, it is unknown to what extent the serum biochemical parameters and hormones differ among these cows, and establishing further knowledge on this subject may help to evaluate the physiological state of dairy cows and to clarify the physiological regulation for variant milk performance traits.

The variations in serum biochemical parameters and hormones under the peripartum, early-lactation, and peak-lactation period have been investigated for years [1,3]. However, the variability of those parameters in mid-lactation cows has not drawn much attention due to their relatively stable physiological states in comparison to the other lactation stages. On the other hand, studies on parity effects have mainly focused on variation analysis between primiparous and multiparous cows [6]. Xue et al. (2018) reported that days-in-milk (DIM) and parity (ranging from 2–7) significantly affected the relative abundance of rumen bacteria in multiparous mid-lactation cows [7]. It is well documented that rumen microbial fermentation products can affect the concentration of metabolites in the blood such as blood urea nitrogen (BUN), glucose, and ketone [8]. Therefore, we hypothesized that DIM and parity could also affect serum biochemical parameters in mid-lactation cows. Serum hormones are usually used to reflect the real-time physiological status of dairy cows [1], so the variation analysis of hormone profiles under different DIM periods or parities can allow physiological shifts to be identified with significantly more precision. 

The objective of this study was to investigate the effects of the DIM and parity on the serum biochemical parameters and hormone profiles of Holstein dairy cows under the same diet and management conditions. 

## 2. Materials and Methods

### 2.1. Animals and Management

All experimental designs and protocols were approved by the Animal Care Committee (ZJU20170422) at Zhejiang University (Hangzhou, P. R. China), and the study was performed following the university’s guidelines (IDS1703) for animal research.

A total of 287 healthy Holstein dairy cows with DIM of 156 (± 33.5 SD) and parity of 3.03 (± 1.12 SD) were selected from the Hangjiang Dairy Farm (Hangzhou, China). Feeding and management of the dairy cows have been described previously [9]. Briefly, the cows were kept individually in well-ventilated tethered stalls in a barn, and had free access to drinking water. They were fed total mixed ration [concentrate: forage = 57: 43, Dry matter (DM) basis] and milked three times per day at 06:30, 14:00, and 20:00 h.

### 2.2. Sample Collection and Measurement

The diet ingredients and chemical compositions were collected and measured and are shown in Appendix A. Milk yields were recorded and milk samples were sampled to determine milk composition. Blood samples were collected using pro-coagulation tubes before morning feeding (04:00–06:00), and serum was obtained for analysis of serum biochemical parameters. The details for collection and treatment of the samples were described previously [9]. The determined serum biochemical parameters include BUN, total protein, albumin, glucose, non-esterified fatty acids (NEFA), total cholesterol, β-hydroxybutyrate (BHB), triglyceride, creatinine, total bilirubin, malonaldehyde (MDA), superoxide dismutase (SOD), glutathione peroxidase (GSH-Px), and total antioxidant capability (T-AOC).

A total of 500 μL of serum was subjected to hormone profiling using colorimetric commercial kits (Jiangsu Meibiao Biological Technology Co. Ltd., Yancheng, China) in a DG5033A microplate reader (Nanjing Huadong Electronics Group Medical Equipment Co. Ltd., Nanjing, China) according to the manufacturer’s protocol. The determined hormones include insulin, glucagon, insulin-like growth factor 1 (IGF-1), cortisol, ghrelin, and leptin. The revised quantitative insulin sensitivity check index (RQUICKI) was calculated using the method reported by Holtenius and Holtenius (2007) [10]. 

### 2.3. Grouping Based on Days-in-Milk and Parity

Experimental cows were divided into four groups based on their DIM: 90–119, 120–149, 150–179, and 180–219, as reported in Xue et al. (2018) [7]. To explore the parity effects, experimental cows were separated into five groups based on the parity range (parity: 2–6). All of the ‘groups’ of cows were housed individually in the same pen to avoid the interference of environment and management.

### 2.4. Statistical Analysis

Statistical analysis was performed using SAS 9.2 (SAS Institute Inc., Cary, NC, USA). Notably, variable outliers were identified as those beyond 3 standard deviations of the mean and were discarded [11]. The 95% confidence interval (CI) was calculated based on the SAS PROC MEANS model. The PROC UNIVARIATE model and PROC DISCRIM model were used to conduct the normality and homogeneity test, respectively. One-way ANOVA or a non-parametric test was utilized to analyze the DIM and parity contributions based on the normality and homogeneity test results. Bonferroni correction was utilized to correct the *p* values of multiple testing among different DIM or parity groups. Two-way ANOVA was performed to assess the interaction of DIM and parity effects. Statistical significance of the main effects was declared at *p* < 0.05, and tendencies were accepted at 0.05 < *p* < 0.10.

## 3. Results and Discussion

### 3.1. General Information

In this study, high individual variations existed in milk yield (coefficient of variation, CV = 17.1%) and milk protein yield (CV = 15.6%). The BUN, glucose, NEFA, total cholesterol, BHB, triglyceride, creatinine, total bilirubin, MDA, and T-AOC also had high variability (CV > 15%). All hormones except for RQUICKI showed a high individual variation (CV > 15%). Notably, although high variability existed in the above parameters, all variations are within the range of homeostatic status as Cozzi et al. (2011) reported [11]. 

The serum biochemical parameters measured from 287 cows could serve as reference information in order to evaluate the metabolic status of multiparous mid-lactation Holstein cows. Although researchers have reported references regarding ruminant serum biochemical parameters, the information was too broad to distinguish the difference between dairy cows and beef cattle [12]. For dairy cows, the available references of blood biochemical parameters were only based on a relatively small cohort (*n* = 20) of dairy cows [13]. Thus, the reference intervals of milk performance and serum biochemical parameters based on the 287 lactating dairy cows are of significant value to provide guidance for metabolic evaluation and nutrition adjustment in the dairy industry.

### 3.2. Effects of Days-in-Milk on Serum Biochemical Parameters and Hormones

The DIM exerted significant effects on milk yield, milk protein content, milk urea nitrogen content (MUN), and somatic cell counts (SCC) (*p* < 0.001, Table 1). It has been well established that milk performance varies through the different lactation stages [5]. However, there is no well-described research on how milk performance traits vary within the whole mid-lactation period, which was covered in this study. No significant difference in parity existed in the four DIM groups.

Concentrations of 10 serum biochemical parameters were significantly different among cows with varied DIM, including the BUN, total protein, glucose, NEFA, BHB, creatinine, total bilirubin, MDA, GSH-Px, and T-AOC (*p* < 0.05, Table 1). As reported by Puppel and Kuczyńska (2016) [14], the changes in BUN, total protein, creatinine, and total bilirubin can be used to evaluate the protein status of dairy cows. The milk protein yield of dairy cows did not change with different DIM (*p* = 0.351), but milk protein content tended to increase with DIM (*p* < 0.001). The higher BUN in the DIM 150–179 and DIM 180–219 groups (4.33 and 4.47 g/L, respectively) compared with the DIM 90–119 and DIM 120–149 groups (3.97 and 3.70 g/L, respectively; *p* < 0.01) may imply that more ammonia was metabolized to urea in the liver at these later DIM stages. Additionally, the lower total protein concentration in the DIM 150–179 and DIM 180–219 groups (62.7 and 59.2 g/L, respectively) than in the DIM 90–119 and DIM 120–149 groups (66.6 and 66.7 g/L, respectively; *p* < 0.01) indicated undesired protein efficiency in dairy cows at the later stage. The overall increasing glucose (*p* = 0.046) and NEFA (*p* = 0.005) concentrations with DIM, and the high BHB concentration in the DIM 90–119 group compared with those in DIM 150–179 and DIM 180–219 groups (*p* = 0.008) reflected the energy status changes in the mid-lactation dairy cows. As an indicator of energy balance, the changes in BHB concentration under different DIM periods align with the milk yield changes, as reported by Sakowski et al. (2012) [15]. Furthermore, the BHB has been reported to upregulate the expression of milk protein synthesis genes in bovine mammary epithelial cells [16]. However, the changes in BHB in this study were opposite to the milk protein variations in the different DIM groups. It warrants further study how the BHB is involved in milk protein synthesis. The dynamic changes in MDA concentration, GSH-Px, and T-AOC activity during the mid-lactation period indicated a fluctuation in oxidative stress related to milk synthesis. In the current study, MDA concentration was significantly higher in the first two DIM groups than that in the latter two groups (5.00, 4.72, 3.29, and 3.64 nmol/L, respectively; *p* < 0.001), which was partly consistent with the milk yield changes. 

The effects of DIM on hormone profiles are shown in Figure 1a. First, glucagon concentration was significantly lower in the DIM 90–119 group than the DIM 120–149 group (88.2, 98.0, 95.2, and 95.2 pg/mL, respectively; *p* = 0.012), which is partly consistent with the change in milk protein content. However, Bobe et al. (2009) found that glucagon infusions decreased milk protein content and yield [17]. The inconsistent results may be derived from different glucagon doses and variant lactation stages, as Hippen et al. (1999) reported [18]. Moreover, both the IGF-1 concentration (*p* = 0.001) and RQUICKI (*p* < 0.001) showed a curvilinear change with increasing DIM. IGF-1 was reported to increase milk yield or milk protein synthesis in in vitro studies [19,20]. However, no corresponding result was found between IGF-1 and milk yield or milk protein content in this study. In our current study, RQUICKI was affected by the DIM periods, as previously reported [21]. The relatively higher RQUICKI in the DIM 90–179 group indicated higher insulin sensitivity to promote the glucose utilization for lactation and maintenance in earlier mid-lactation, which was proved by the relatively lower glucose concentration and higher milk yield in those DIM periods.

### 3.3. Effects of Parity on Serum Biochemical Parameters and Hormones

Milk yield, milk protein content and yield, MUN, and SCC were affected by parity (*p* < 0.05, Table 2). Similarly, research related to the effects of parity on milk performance traits has been studied for years [5], and was not the focus of this research. No significant difference existed in DIM within any parity group (*p* > 0.05).

In the current study, the concentrations of five serum biochemical parameters were affected by parity (*p* < 0.05, Table 2). Serum total protein concentration was higher in dairy cows with parity 5 than those with parity 2 (*p* = 0.005, Table 2), which aligns with the findings of Blum et al. (1983) [22]. The serum total cholesterol concentration was higher in cows with parity 2 than those with parity 3 and parity 6 (6.35, 5.86, 5.87, 5.81, and 5.24 mmol/L for parities 2 to 6, respectively; *p* = 0.001), which might reveal more lipid mobilization to provide energy for more milk production in parity-2 cows. Serum BHB concentration was relatively stable in cows with parities 2 to 5, but lower in parity-6 cows (0.50, 0.51, 0.49, 0.47, and 0.40 mmol/L for parities 2 to 6, respectively; *p* = 0.035). Under different parities, the consistent changes in the BHB concentration and milk yield revealed the significance of energy supply for milk synthesis. Creatinine concentration in the serum was highest in dairy cows with parity 2 and decreased until parity 5 (62.5, 59.1, 57.2, 54.3, and 58.5 μmol/L for parities 2 to 6, respectively; *p* = 0.004). The SOD concentration declined with increasing parity (*p* = 0.034). For all serum biochemical parameters, no interaction between DIM and parity effects was observed (*p* > 0.05).

All hormones that we detected were kept stable under the different parities (*p* > 0.05, Figure 1b). However, an interaction effect of DIM and parity on glucagon concentration (*p* = 0.015) was detected. No parity effect on the ghrelin level is in accordance with Honig et al. (2016), [23], which reflects that feed intake was enough to provide energy for lactation, and energy homeostasis was not the primary cause for variant milk performance traits of these experimental cows under different parity groups. The above results suggest that the hormones involved in energy and appetite are not the critical endocrine mediators to the observed variations in lactation performance with parity. 

## 4. Conclusions

In conclusion, DIM and parity contribute to the variations in serum biochemical parameters and hormones related to protein status, energy supply, liver and kidney function, and oxidative stress of mid-lactation Holstein dairy cows, with the effect of DIM being dominant over parity. For multiparous mid-lactation cows, DIM and parity should also be considered to assist in optimizing the nutrition and management strategies according to their physiological metabolism.

## Figures and Tables

**Figure 1 animals-09-00230-f001:**
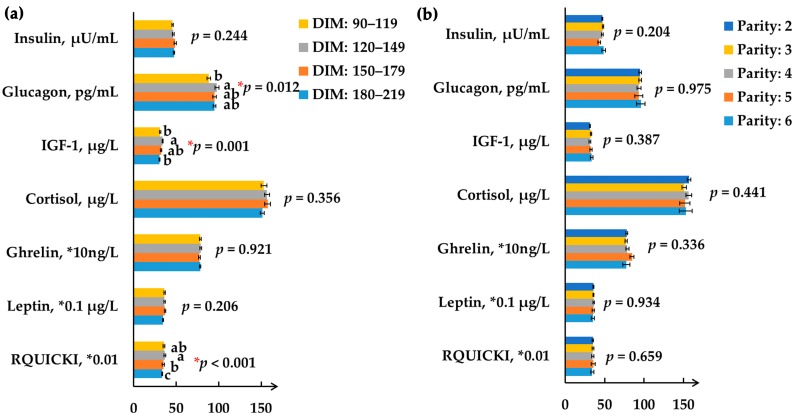
Effect of days-in-milk (**a**) and parity (**b**) on serum hormone profiles. (**a**) Description of days-in-milk effects on serum hormone profiles; (**b**) Description of parity effects on serum hormone profiles. Different color bars represent varying days-in-milk or different parities. Bar height represents the mean value of each individual hormone; the error bars represent the standard error of mean. IGF-1: insulin-like growth factor 1; RQUICKI: revised quantitative insulin sensitivity check index [10]. ^a,b^ Means within a hormone with different superscripts significantly differ (*p* < 0.05).

**Table 1 animals-09-00230-t001:** Effects of days-in-milk on milk performance and blood biochemical parameters.

Item	Days in Milk				SEM ^1^	*p*-value
	90–119 (*n* = 53)	120–149 (*n* = 74)	150–179 (*n* = 57)	180–219 (*n* = 103)		
Milk performance						
Milk yield, kg/d	36.0 ^a^	33.9 ^ab^	31.7 ^b^	32.4 ^b^	0.336	<0.001
Milk protein, %	2.89 ^b^	3.08 ^a^	3.13 ^a^	3.14 ^a^	0.015	<0.001
Milk protein yield, kg/d	1.04	1.04	0.99	1.01	0.009	0.351
Milk urea nitrogen, mg/dL	13.4 ^c^	13.8 ^bc^	14.5 ^ab^	14.8 ^a^	0.116	<0.001
Somatic cell counts, ×10^3^/mL	170 ^bc^	168 ^c^	284 ^a^	244 ^ab^	14.474	<0.001
Serum biochemical parameters						
Blood urea nitrogen, mmol/L	3.97 ^b^	3.70 ^c^	4.33 ^a^	4.47 ^a^	0.049	<0.001
Total protein, g/L	66.6 ^a^	66.7 ^a^	62.7 ^b^	59.2 ^c^	0.440	<0.001
Albumin, g/L	37.7	39.3	37.6	38.8	0.303	0.190
Glucose, mmol/L	3.94 ^c^	4.28 ^b^	4.49 ^ab^	4.73 ^a^	0.046	<0.001
NEFA, mmol/L ^2^	0.22 ^b^	0.17 ^c^	0.21 ^b^	0.26 ^a^	0.005	<0.001
Total cholesterol, mmol/L	6.01	6.09	5.71	6.18	0.067	0.090
β-hydroxybutyrate, mmol/L	0.55 ^a^	0.49 ^ab^	0.46 ^b^	0.49 ^b^	0.008	0.006
Triglyceride, mmol/L	0.29	0.31	0.29	0.29	0.005	0.354
Creatinine, μmol/L	56.4 ^b^	56.7 ^b^	62.2 ^a^	62.5 ^a^	0.685	<0.001
Total bilirubin, μmol/L	3.92 ^b^	4.08 ^b^	4.71 ^a^	4.93 ^a^	0.061	<0.001
Malonaldehyde, nmol/mL	5.00 ^a^	4.72 ^a^	3.29 ^b^	3.64 ^b^	0.091	<0.001
Superoxide dismutase, U/mL	136	135	136	136	0.796	0.833
Glutathione peroxidase, U/mL	128 ^ab^	131 ^ab^	134 ^a^	126 ^b^	1.098	0.038
T-AOC, U/mL ^3^	4.01 ^a^	3.52 ^b^	3.42 ^b^	3.54 ^b^	0.050	0.001
Parity	2.92	3.16	3.00	3.01	0.066	0.619

^1^ SEM: Standard error of mean. ^2^ NEFA: Non-esterified fatty acids. ^3^ T-AOC: Total antioxidant capability. ^a,b,c^ Means within a row with different superscripts significantly differ (*p* < 0.05).

**Table 2 animals-09-00230-t002:** Effects of parity on milk performance and blood biochemical parameters.

Item	Parity					SEM ^1^	*p*-value
	2 (*n* = 117)	3 (*n* = 89)	4 (*n* = 49)	5 (*n* = 19)	6 (*n* = 13)		
Milk performance							
Milk yield, kg/d	34.0 ^a^	33.6 ^a^	32.9 ^ab^	31.9 ^ab^	28.9 ^b^	0.336	0.020
Milk protein, %	3.12 ^a^	3.09 ^ab^	3.02 ^abc^	2.98 ^bc^	2.94 ^c^	0.015	0.006
Milk protein yield, kg/d	1.06 ^a^	1.03 ^ab^	0.98 ^abc^	0.94 ^bc^	0.85 ^c^	0.009	0.020
Milk urea nitrogen, mg/dL	14.7 ^a^	14.0 ^ab^	14.1 ^ab^	14.1 ^ab^	13.0 ^b^	0.116	0.007
Somatic cell counts, ×10^3^/mL	152 ^b^	234 ^ab^	289 ^a^	296 ^ab^	341 ^a^	14.474	0.002
Serum biochemical parameters							
Blood urea nitrogen, mmol/L	4.13	4.11	4.32	4.02	4.03	0.049	0.544
Total protein, g/L	61.6 ^b^	63.8 ^ab^	63.4 ^ab^	67.5 ^a^	66.4 ^ab^	0.440	0.005
Albumin, g/L	38.9	38.5	38.7	36.4	36.8	0.303	0.251
Glucose, mmol/L	4.30	4.51	4.54	4.36	4.50	0.046	0.237
NEFA, mmol/L ^2^	0.22	0.21	0.23	0.22	0.23	0.005	0.837
Total cholesterol, mmol/L	6.35 ^a^	5.86 ^b^	5.87 ^ab^	5.81 ^ab^	5.24 ^b^	0.067	0.001
β-hydroxybutyrate, mmol/L	0.50 ^ab^	0.51 ^a^	0.49 ^ab^	0.47 ^ab^	0.40 ^b^	0.008	0.035
Triglyceride, mmol/L	0.29	0.30	0.30	0.31	0.32	0.005	0.595
Creatinine, μmol/L	62.5 ^a^	59.1 ^ab^	57.2 ^ab^	54.3 ^b^	58.5 ^ab^	0.685	0.008
Total bilirubin, μmol/L	4.43	4.62	4.28	4.47	4.20	0.061	0.584
Malonaldehyde, nmol/mL	4.19	4.15	3.83	4.00	4.07	0.091	0.664
Superoxide dismutase, U/mL	138 ^a^	135 ^ab^	133 ^ab^	131 ^b^	131 ^b^	0.796	0.034
Glutathione peroxidase, U/mL	127	131	128	133	128	1.098	0.487
T-AOC, U/mL ^3^	3.6	3.57	3.72	3.36	3.63	0.050	0.663
Days in milk	153	161	160	158	139	1.979	0.481

^1^ SEM: Standard error of mean. ^2^ NEFA: Non-esterified fatty acids. ^3^ T-AOC: Total antioxidant capability. ^a,b^ Means within a row with different superscripts significantly differ (*p* < 0.05).

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
