# Peer review of "Days-in-Milk and Parity Affected Serum Biochemical Parameters and Hormone Profiles in Mid-Lactation Holstein Cows"

_animals, 2019, doi:10.3390/ani9050230_

Round 1
Reviewer 1 Report
Overall this manuscript describes an interesting study assessing the impact of parity and days in milk (DIM) on various blood parameters in dairy cows. The paper is generally well-written and needs only moderate revisions before being acceptable for publication.
Line 23. This part is not in the results/statistics, suggest removing from the summary'
Line 29. Can authors add a short description of the background to their research before the start of the abstract
Line 39. Can authors add details to the direction to the effect of this interaction?
Line 40. How do authors know that the DIM is more 'predominant' than parity?
Line 78 and 82. Although this is presented elsewhere, can authors also add a brief description of the methods here?
Line 94. Can authors add more description about how the 'groups' of cows were housed? Were cows actually separated in the 'groups' in different pens based on their DIM/parity? Or were cows/groups housed together in the same pens?
Line 104. Was anything else included in the models?
Line 105. Authors do a lot of tests here which may impact their ability to accurately detect a P<0.05 that is not by random chance. Did authors consider doing a bonferroni correction to their P-value to account for experimentwise error?
Figure 1. The font on this figure is too small to be readable - suggest making larger.
Author Response
Point 1:
Comments and Suggestions for Authors
Overall this manuscript describes an interesting study assessing the impact of parity and days in milk (DIM) on various blood parameters in dairy cows. The paper is generally well-written and needs only moderate revisions before being acceptable for publication.
Response 1: Thanks for your encouragement. We have revised the manuscript as you suggested.
Point 2: Line 23. This part is not in the results/statistics, suggest removing from the summary'
Response 2: Removed as you suggested. Besides, the corresponding sentences in Line 69-70 have also been removed.
Point 3: Line 29. Can authors add a short description of the background to their research before the start of the abstract.
Response 3: Revised as you suggested.
Point 4: Line 39. Can authors add details to the direction to the effect of this interaction?
Response 4: Revised as you suggested.
Point 5: Line 40. How do authors know that the DIM is more 'predominant' than parity?
Response 5: We’re sorry to make you confused with this sentence. We have revised the sentence.
Point 6: Line 78 and 82. Although this is presented elsewhere, can authors also add a brief description of the methods here?
Response 6: Following your suggestion, a brief description of the method was added.
Point 7: Line 94. Can authors add more description about how the 'groups' of cows were housed? Were cows actually separated in the 'groups' in different pens based on their DIM/parity? Or were cows/groups housed together in the same pens?
Response 7: Thanks for your question. All of the ‘groups’ of cows were housed individually in the same pen to avoid the interference of environment and management.
Point 8: Line 104. Was anything else included in the models?
Response 8: Thanks for your question. The diet, management and environment of all experimental cows were controlled to be identical, so we did not include other factors in the models.
Point 9: Line 105. Authors do a lot of tests here which may impact their ability to accurately detect a P<0.05 that is not by random chance. Did authors consider doing a bonferroni correction to their P-value to account for experiment wise error?
Response 9: Thanks for your suggestion. We did the bonferroni correction as you suggested, and the corresponding results were updated.
Point 10: Figure 1. The font on this figure is too small to be readable - suggest making larger.
Response 10: Revised as you suggested.

Reviewer 2 Report
I cannot understand importance of your study.
You have already published
9. Wu, X.; Sun, H.; Xue, M.; Wang, D.; Guan, L.L.; Liu, J. Serum metabolome profiling revealed potential biomarkers for milk protein yield in dairy cows. Journal of Proteomics 2018, 184, 54-61; DOI:10.1016/j.jprot.2018.06.005.
You are trying one more manuscript with remaining data of [9].
Serum biochemical parameters and hormones directly reflect the physiological state of dairy cows and balance between nutrient intake and milk yield.
High intake - high milk --- good or bad parameters
High intake - low milk --- good parameters
Low intake - high milk --- bad parameters
Low intake - low milk --- good or bad parameters
We already know lactation curves and effects of parity for milk yield.
What is new ?
L54; However, the variability of those parameters in mid-lactation cows have not drawn much attention due to the supposed stable physiological state. --- Why do you suppose drastic change of physiological state in mid-lactation cows ?
L57; There are few studies focused on serum biochemical and hormonal differences among multiparous cows. --- Really ? Are you the first scientist in the world ?
L67; investigate the effects of the DIM and parity on serum biochemical parameters and hormone profiles in Holstein dairy cows under the same nutrition and management conditions. --- Could you really control Holstein dairy cows under the same nutrition and management conditions ? Where are data of nutrient intake ?
L136; can be used to evaluate protein status of dairy cows --- Milk protein yield, kg/d did not changed (1.04 1.04 0.99 1.01). How do you think about this ?
L140; more ammonia ... undesired protein efficiency in dairy cows at this stage --- What do you think this even under the same nutrition and management conditions ?
L144; decreased globulin content --- Where are data ?
L145; reduced lactating stress with DIM --- Do you have any proofs about this ? Did you observe reduced stress ?
L147; reflected the energy status changes in the mid-lactation dairy cows --- I have already known/ What is new ? Do you say only energy status changes ? No tendency ? New law ?
L153; However, in this study, the changes in BHB ... inconsistent changes. --- What do you want to say finally ?
L207; feed intake was enough to provide energy for lactation --- This is all. How do you think ?
L214; The changes in serum biochemical parameters and hormones within multiparous mid-lactation dairy cows emphasize the significance of taking the DIM periods and parity into consideration in future studies to precisely reflect the physiological metabolism of dairy cows. --- What is new information ? Do you think we a lot of researcher of Animal Science in the world have never known about this till now ?
Table 1. 180-219 (n=103) --- Why did you use a lot of cows only in this period ? I think biased data.
Why did not observe milk fat ? I think it is very important for dairy industry.
Table 2. Parity 5 (n=19) Parity 6 (n=13) --- Why did you use few cows only in these periods ? I think biased data. If you slaughtered a lot of young cows (Parity 2,3,4), it is biased data.
Author Response
Point 1:
Comments and Suggestions for Authors
I cannot understand importance of your study.
You have already published
9. Wu, X.; Sun, H.; Xue, M.; Wang, D.; Guan, L.L.; Liu, J. Serum metabolome profiling revealed potential biomarkers for milk protein yield in dairy cows. Journal of Proteomics 2018, 184, 54-61; DOI:10.1016/j.jprot.2018.06.005.
You are trying one more manuscript with remaining data of [9].
Response 1: Thanks for your comments and question.
The objectives of these two papers were not the same. This manuscript was focus on the DIM and parity effects on serum biochemical parameters and hormone profiles, which was designed to get the physiological changes of dairy cows with DIM / parity. Besides, we integrated the changes of milk performance traits with serum biochemical parameters and hormones to identify physiological indictors for milk production traits. However, the previous manuscript (Wu et al., 2018) was to identify the metabolic mechanisms and biomarkers relating to milk protein yield, which was dependent on both milk yield and milk protein content.
This manuscript was designed based on a relative larger cohort of dairy cows (n = 287), which should be valuable to provide reference information to evaluate the metabolic status of multiparous mid-lactation Holstein cows. And we added the hormone information in this study to reflect the real-time physiological status of dairy cows, which was not included in our previous study.
Point 2: Serum biochemical parameters and hormones directly reflect the physiological state of dairy cows and balance between nutrient intake and milk yield.
High intake - high milk --- good or bad parameters
High intake - low milk --- good parameters
Low intake - high milk --- bad parameters
Low intake - low milk --- good or bad parameters
We already know lactation curves and effects of parity for milk yield.
What is new?
Response 2: Thanks for your question. It is true that serum biochemical parameters and hormones directly reflect the physiological state of dairy cows and balance between nutrient intake and milk yield. The lactation stage and parity contribute to the milk yield of dairy cow, however, in this manuscript, we focused on the effects of DIM and parity on the physiological and metabolic status of dairy cows, which could be reflective of some serum biochemical parameters and hormones. In other word, this study was not only for clarifying the physiological roles for variant milk performance traits, but also help us to understand the changes of physiological and metabolic status of dairy cows with parity and DIM, which may provide potential strategies for a better performance.
Point 3: L54; However, the variability of those parameters in mid-lactation cows has not drawn much attention due to the supposed stable physiological state. --- Why do you suppose drastic change of physiological state in mid-lactation cows?
Response 3: We are sorry to express our meaning in a confused manner. We hope to state here that due to relatively stable physiological state at the mid-lactation than at other stages, the variability of these serum parameters in mid-lactation cows have not drawn much attention. We have revised the sentence in the revised manuscript.
Point 4: L57; There are few studies focused on serum biochemical and hormonal differences among multiparous cows. --- Really? Are you the first scientist in the world?
Response 4: We are sorry for the bad expression. Of course we are not the first scientist in this area. We tried to convey that only a small proportion of studies have been focused on the comparison among multiparous cows, especially relating to the hormone profiles. To avoid confusion and mistake, we deleted this sentence in the manuscript.
Point 5:
L67; investigate the effects of the DIM and parity on serum biochemical parameters and hormone profiles in Holstein dairy cows under the same nutrition and management conditions. --- Could you really control Holstein dairy cows under the same nutrition and management conditions? Where are data of nutrient intake?
Response 5: Thanks for your question. We did not exactly control the same nutrition and management conditions, but we fed all the cows with same diet under identical management. We are sorry we did not have data of nutrient intake.
Point 6: L136; can be used to evaluate protein status of dairy cows --- Milk protein yield, kg/d did not change (1.04 1.04 0.99 1.01). How do you think about this?
Response 6: Thanks for your question. Milk protein yield of dairy cows did not change with different DIM, but milk protein content tended to increase with DIM. The sentence was revised.
Point 7: L140; more ammonia ... undesired protein efficiency in dairy cows at this stage --- What do you think this even under the same nutrition and management conditions?
Response 7: Thanks for your question. We consider that higher BUN in the DIM 150-179 and DIM 180-219 groups may be attributed to more ammonia to be metabolized to urea. Because the cows were fed with same diet under identical management, these differences were owing to the difference in DIM. The sentence was reworded.
Point 8: L144; decreased globulin content --- Where are data?
Response 8: Sorry for the confusion. The sentence was reworded.
Point 9: L145; reduced lactating stress with DIM --- Do you have any proofs about this? Did you observe reduced stress?
Response 9: Similar to the above point 8, the sentence was reworded.
Point 10: L147; reflected the energy status changes in the mid-lactation dairy cows --- I have already known/ What is new? Do you say only energy status changes? No tendency? New law?
Response 10: Thanks for your question. These serum variables can be used to reflect energy status. Here we like to indicate the change in energy status with DIM based on the change in these variables.
Point 11: L153; However, in this study, the changes in BHB ... inconsistent changes. --- What do you want to say finally?
Response 11: Thanks for your question. We plan to state the inconsistency between our study and Wang et al. [Ref. 16] may be attributed to the biological complexity in the in-vivo study. The sentence was reworded.
Point 12: L207; feed intake was enough to provide energy for lactation --- This is all. How do you think?
Response 12: Thanks for your question. We inferred that the feed intake was not the main cause for variant milk performance traits. Other factors like feed efficiency might contribute to the different milk performance traits at different DIM or parity.
Point 13: L214; The changes in serum biochemical parameters and hormones within multiparous mid-lactation dairy cows emphasize the significance of taking the DIM periods and parity into consideration in future studies to precisely reflect the physiological metabolism of dairy cows. --- What is new information? Do you think we a lot of researcher of Animal Science in the world have never known about this till now?
Response 13: Thanks for your critical comments. To avoid confusion and misunderstanding, we reworded the sentence.
Point 14: Table 1. 180-219 (n=103) --- Why did you use a lot of cows only in this period? I think biased data.
Why did not observe milk fat? I think it is very important for dairy industry.
Response 14: Thanks for your question. For the first question, it depends on the distribution of dairy cows in the experimental dairy farm. All of the health multiparous mid-lactation cows we could find in this farm have been selected in this study.
You are right that the milk fat data was important for dairy industry. However, our study focused on the milk protein, so we did not mention the milk fat in this study.
Point 15: Table 2. Parity 5 (n=19) Parity 6 (n=13) --- Why did you use few cows only in these periods? I think biased data. If you slaughtered a lot of young cows (Parity 2,3,4), it is biased data.
Response 15: Thanks for your question. As we mentioned before, the number of cows depends on the distribution of dairy cows in the experimental farm. To keep the high profit, Chinese dairy farms do not keep their cows at high parity. Therefore, less cows were obtained in Parity 5 and 6.
